# Highly Luminescent Crystalline Sponge: Sensing Properties and Direct X-ray Visualization of the Substrates

**DOI:** 10.3390/molecules27228055

**Published:** 2022-11-19

**Authors:** Pavel A. Demakov, Alexey A. Ryadun, Danil N. Dybtsev

**Affiliations:** Nikolaev Institute of Inorganic Chemistry, Siberian Branch of the Russian Academy of Sciences, 3 Akad. Lavrentieva Ave., 630090 Novosibirsk, Russia

**Keywords:** metal-organic frameworks, coordination polymers, crystal structure, host-guest interactions, flexibility, luminescence, sensing

## Abstract

A phenomenon of crystalline sponge is represented by guest-dependent structural fluidity of the host polymeric lattice in highly crystalline sorbents, such as metal-organic frameworks, driven by multiple weak intermolecular interactions. Such induced fitting in MOFs is a valuable property in selective adsorption, guest determination by single-crystal XRD and in-situ structural analysis under external stimuli. In this work, a porous three-dimensional metal-organic framework [Eu_2_(DMF)_4_(ttdc)_3_]·4.45DMF (**1_DMF_**; DMF = *N*,*N*-dimethylformamide, ttdc^2–^ = trans-thienothiophenedicarboxylate anion) was applied as a crystalline sponge bearing luminescent functionality to couple its sensing properties with direct structural determination of the adsorbed molecules. As a result, the paper discusses crystal structures and luminescent properties for the successfully obtained new adducts with the crystallographic formulae [Eu_2_(DMSO)_4_(ttdc)_3_]·2.5DMSO·2.2H_2_O (**1_DMSO_**; DMSO = dimethylsulfoxide), [Eu_2_(DMF)_4_(ttdc)_3_]·3phet (**1_phet_**; phet = phenylethanal) and [Eu_2_(DMF)_3.5_(cin)_0.5_(ttdc)_3_]·1.64cin (**1_cin_**; cin = *trans*-cinnamaldehyde). As a result of inclusion of DMSO into **1**, a slight increase in the quantum yield and excited state phosphorescence lifetime was observed, while the adsorption of phet leads to a considerable (up to three times) decrease in the corresponding values. The incorporation of cinnamal results in a full quenching of QY, from 20% down to zero, and a more than order of magnitude diminishing of the excited state lifetime compared to the initial **1_DMF_**. The effective sensing of cinnamal was explained from the structural point of view by its direct coordination to the Eu^3+^ emitter, as well as by multiple weak intermolecular interactions with ttdc antenna ligand, both capable of enhancing the non-radiative energy dissipation.

## 1. Introduction

The crystalline sponge method represents a convenient route for the single-crystal X-ray structural analysis of the molecules with a high degree of flexibility, which are hard to crystallize in a pure form [1,2,3,4,5,6]. Binding such guests into a porous host by multiple intermolecular interactions allows the molecular mobility of the adsorbed moieties to diminish, providing a necessary basis for their direct structure determination. The application of crystalline sponges in chiral and natural compounds analysis [7,8,9], chromatograph detectors [10,11] and mechanistic studies [12,13,14,15,16,17,18] has emerged in recent years. Flexible metal-organic frameworks (MOFs) are recognized as valuable hosts for such method [8,19,20,21] due to the possibility of retaining a high crystallinity, even at a pronounced breathing. For example, an up to four times variation of unit cell parameters has been reported for MIL-88 series upon activation or inclusion of water and light alcohols [22,23]. In-situ X-ray diffraction methods have been successfully applied in the structural determination of three [24] or five [25] forms of MOF crystalline sponge upon single gas adsorption. High tunability of chemical affinity and properties of such mobile coordination frameworks unveil their further applications in selective adsorption, sensing and data storage [26,27,28,29,30,31,32,33], coupled with facilitated X-ray structural analysis of the incorporated guests.

Previously, a crystal structure of a compound [Eu_2_(DMF)_4_(ttdc)_3_]·4DMF (**1_DMF_**; DMF = *N*,*N*-dimethylformamide, H_2_ttdc = thieno[3,2b]thiophene-2,5-dicarboxylic acid) was determined by our group at T = 140 K [34]. This MOF possesses a three-dimensional, highly porous coordination framework bearing primitive cubic *pcu* topology, which is known for a pronounced breathing by metal node deformation, rigid linker bending or flexible linker rotation mechanisms. Highly effective aromatic antenna-driven luminescence of Eu^3+^ was revealed for **1_DMF_**, as well as a sensing potential of this system was unveiled for either this compound [34] or its isostructural terbium(III)-based analogue [35]. Having such a combination of potential structural dynamics and luminescent sensing abilities, we decided to further investigate a solvent exchange in this MOF using a combination of single-crystal X-ray diffraction analysis and luminescence measurements. As a result, this work reports crystal structures and luminescent properties of new adducts of **1** with dimethylsulfoxide (**1_DMSO_**), phenylethanal (**1_phet_**) and cinnamal (**1_cin_**; Figure 1). In particular, very strong phosphorescence quenching from 20% quantum yield down to zero, with more than a 10-fold decrease in an excited state lifetime, was revealed for **1_cin_**, which was explained from the structural point of view.

## 2. Results and Discussion

### 2.1. Crystal Structures

Adducts **1_DMSO_**, **1_cin_** and **1_phet_** were obtained by the immersion of the **1_DMF_** in the corresponding liquids. Similarly to the initial **1_DMF_** structure, Eu^3+^ adopts a distorted square antiprismatic geometry with coordination number (CN) = 9 in all the structures. Two O atoms of the coordinated solvents and two O atoms of the κ^2^-COO group represent a prism base, while two O atoms of the κ^1^,κ^1^-COO group and two O atoms of the chelating κ^2^,κ^1^-COO group represent an upper plane of the prism. An O atom of the non-chelating κ^2^,κ^1^-COO group acts as a cap at the upper plane, filling the CN of the metal cation to 9. Bond lengths of the Eu(III) coordination environment in all the adducts are summarized in Appendix A and fit the typical values for Eu(III) and its nearest neighbors among the lanthanide(III) row in cognate structures [36,37,38,39,40,41]. Two neighboring metal ions couple into a binuclear carboxylate building blocks with the general formulae {Eu_2_(O_solv_)_4_(OOCR-κ^2^)_2_(μ-OOCR-κ^1^,κ^1^)_2_(μ-OOCR-κ^2^,κ^1^)_2_} (Figure 1a–c). Therefore, the connectivity in the six-connected binuclear metal node fully retains at solvent substitution. Figure 1d shows a schematical illustration of the cage within a distorted pcu-type Eu-ttdc polymeric lattice, in which each binuclear carboxylate block and ttdc ligand are considered as a node and an edge, respectively. In such presentation, one cage constructed by eight closest metal blocks obviously presents a parallelogram and angles between its independent faces in **1_DMF_** are 49°, 67° and 81°. Differences in coordination ability as well as molecular size of the solvents result in different types of their incorporation into a three-dimensional coordination framework and some local distortions in bending angles, which causes a considerable shifting in the structural parameters of the adducts, illustrated in Table 1.

In the **1_DMSO_** structure, a full substitution of the coordinated DMF by DMSO is observed. As mentioned above, the geometry of Eu^3+^ and the coordination bond lengths in the adducts are similar, except some increase in the distance between metal ion and cap oxygen of the κ^2^,κ^1^-COO group in **1_DMSO_** on ca. 0.14 Å. Such weakening of the coordination bond with the ninth donor atom is apparently related to a higher donor ability of DMSO ligand compared to DMF, which will be shown below to influence the luminescent properties of the **1_DMSO_** adduct. Parallelogram angles (Appendix A) of the cage in the **1_DMSO_** structure are 54°, 64° and 74°, indicating some straightening of the cage shape compared to **1_DMF_**, mainly due to the ca. 10° bending of the κ^2^,κ^1^-ttdc ligand (Appendix A). The solvent accessible volume in the [Eu_2_(DMSO)_4_(ttdc)_3_] coordination framework is 32%, showing a considerable decrease compared to the DMF-coordinated lattice. Only one DMSO and two disordered water positions were determined directly, resulting in one DMSO and one water molecule localized per formula unit. A non-ordered electron density in the residual voids was analyzed using the PLATON/SQUEEZE procedure [43] (75 e^–^ in 254 Å^3^) and attributed to additional 1.5 DMSO and 1.2 H_2_O molecules according to the CHN, IR and TG data, resulting in [Eu_2_(DMSO)_4_(ttdc)_3_]·2.5DMSO·2.2H_2_O as a summary formula of the crystal.

In the **1_cin_** adduct, a partial substitution of the coordinated DMF by cinnamal is observed by 0.5 cinnamal molecule per binuclear carboxylate block, resulting in the formula [Eu_2_(DMF)_3.5_(cin)_0.5_(ttdc)_3_] for the coordination network. An incomplete substitution can be explained by the cinnamal’s poorer coordination ability, compared to DMSO, as well as by the sterical hindrance of the cinnamal ligand possessing a substantially larger molecular size. An overall localization of the cinnamal molecules in the cage of **1_cin_** is presented in Appendix A. An analysis of the intermolecular host-guest interactions was performed to rationalize the positions of the guest cinnamal molecules in the pores. As shown in Figure 2a, a 2.40 Å H31M…O1G distance represents a shortest contact of the first guest cinnamal molecule, suggesting weak hydrogen contacts between the carbonyl H_DMF_ acceptor and O_cinnamal_ donor to be main factor driving the incorporation of cinnamal molecule. Further, the ttdc ligand is involved in the adsorption through H8G…O11 (2.59 Å) contact between the CH fragment of cinnamal aromatic ring and carboxylate oxygen, and H3G…S1 (2.92 Å) contact between the CH fragment of the cinnamal double bond and sulfur atom of the ligand heterocyclic core, showing a significant role of the extended thienothiophene-containing dicarboxylate bridge in the localization of guest molecule on its position. Similar analysis for the second guest molecule (Figure 2b) reveals 2.51 Å H2G…O1’ contact between the CH fragment of the first guest double bond and the second guest oxygen, as well as C15M,C14M,C19M…C11’,C12’,C13’ π-π stacking (3.35 Å as a closest distance and 3.48 Å as an average interplanar distance) in an offset manner, with the conjugated π-system of the coordinated cinnamal, suggesting that the adsorption of the second guest molecule is mostly cinnamal-driven. However, 2.58 Å H16…O22 contact between the CH fragment of the cinnamal aromatic ring and ttdc carboxylic group also exists. Figure 2b also shows the CH…O (2.36 Å and 2.50 Å) interactions between the coordinated cinnamal (shown grey) double bonds and ttdc carboxylic group, and CH-π interactions (2.93 Å average distance) between the CH fragment of the coordinated cinnamal and other ttdc ligand heterocyclic core.

The observed involvement of the ttdc ligand into the adsorption of cinnamal, as well as the presence of a complicated system of both multiple host-guest contacts and interactions between the coordinated cinnamal and ttdc bridges, may strongly impact the structural dynamics of the overall coordination framework, if such dynamics are possible. Indeed, Figure 3 presents the projections of the Eu-ttdc lattices in **1_DMF_** and **1_cin_** along *a* crystallographic axis, showing a ca. 15° bending of the κ^2^-ttdc ligand in **1_cin_**, which results in an up to 2 Å contraction in half of the interlayer distances. The successful localization of the cinnamal guests only in the contracted interlayer space, as well as the absence of ordered guest molecules in another half of the interlayer distances being retained similar to **1_DMF_**, presents an additional confirmation of the deep impact of the cinnamal chemical nature on the observed structural transition in the Eu-ttdc coordination framework. As there are no obvious restrictions on the ttdc bending angle in a decently porous lattice, such type of structural dynamics can be recognized as sponge-like behavior of the host, described in the introduction.

In summary, two different guest cinnamal positions, both with 0.5 occupancies, were determined directly, resulting in one localized C_9_H_8_O molecule per formula unit. A non-ordered electron density in the residual voids (90 e^–^ in 379 Å^3^ per unit cell) was attributed to 0.64 additional cinnamal molecule per formula unit, giving [Eu_2_(DMF)_3.5_(cin)_0.5_(ttdc)_3_]·1.64cin as a general formula of the crystal.

In contrast to the cinnamal-containing adduct, no substitution of DMF by phenylethanal was observed in the Eu^3+^ coordination sphere. Thus, the structural parameters of the coordination framework in **1_phet_**, such as void volume and V/Z (see Table 1), are reasonably similar to the initial **1_DMF_** characteristics. The guest phenylethanal molecules were fully localized directly (Appendix A), with 3C_8_H_8_O total amount per formula unit. It should be noted that no crystal structure of free phenylethanal was reported, and only one structure containing uncoordinated phenylethanal is reported at present [44], in which a conformationally mobile substrate was successfully immobilized in the pores of the crystalline sponge.

An analysis of intermolecular host-guest interactions was also performed for **1_phet_**. As shown in Figure 4, for one C_8_H_8_O molecule, a 2.42 Å H1M…O2G distance is observed, representing weak hydrogen contacts between the carbonyl group of the coordinated DMF and the oxygen atom of phenylethanal. This position mostly repeats the corresponding cinnamal guest localization. For another C_8_H_8_O molecule, which is disordered over two close positions, a short contact between the CH fragment of the ttdc heterocyclic ring and phenylethanal oxygen is observed, with 2.42 Å H15…O1G as the shortest distance, showing phenylethanal to be a donor in weak hydrogen bonding to the thiophene ring acceptor. As the interactions of cinnamal with the heterocyclic core of the ttdc ligand are presented by H_guest_…C_ligand_ contacts (see above), suggesting the heterocycle to be a donor, we can state the inversion of donor/acceptor roles of the thiophene ring upon the inclusion of cinnamal or phenylethanal, having a close chemical nature. Such an unusual observation shows an “ampholytic” behavior of the thiophene ring in the ttdc ligand, capable of forming diverse-type host-guest interactions, depending on the minor structural and electron features of the incorporated guest molecules. Guest-guest bonding in **1_phet_** is represented by CH…π contacts, with 2.53 Å H15G…C2XG_plane_ as the shortest distance.

### 2.2. Characterization

The CHN and TG data for the obtained adducts correspond well to their crystal structures, although some variations in the chemical compositions are observed due to high coordination framework mobility. A guest content in **1_cin_**, obtained by single crystal XRD, is apparently underestimated, as some cinnamal guest molecules were successfully refined only with partial occupancies, without any observable competing positions to append the sum occupancies up to 1.0. According to TG (Appendix A), all the adducts lose the guest and coordinated solvents until ca. 320 °C. Decomposition of the Eu-ttdc lattice in all adducts occurs at ca. 460 °C, indicating the preservation of the main host structural motif during the solvent exchange. All the IR spectra (Appendix A) of the obtained adducts contain typical bands of C(sp^2^)–H vibrations in the aromatic ring, antisymmetric and symmetric coordinated COO group stretchings. In addition, the IR spectrum of **1_DMSO_** contains a very intensive band, at 1013 cm^–1^, corresponding to the characteristic S=O valence vibration in DMSO, and a wide band of O–H valence vibrations, which indicates a presence of a considerable amount of water, thus confirming the X-ray data. The IR spectrum of **1_phet_** contains a band at 1718 cm^–1^, corresponding to the characteristic aldehyde C=O valence stretchings, as well as a wide band of O–H valence vibrations, consistent to a presence of water, also observed by CHN and TG analyses, which possibly indicates some inclusion of the guest H_2_O molecules. The IR spectrum of **1_cin_** contains a band of 1673 cm^–1^, corresponding to the cinnamal conjugated C=O valence vibration [45,46]. In addition, the wide O–H valence band on **1_cin_** spectrum is very weak, which confirms the absence of water in this sample, which is again consistent with the crystal structure and other analysis data.

### 2.3. Luminescent Properties

The solid-state luminescent properties were investigated for the obtained adducts. The excitation spectra of the named compounds are shown in Appendix A. The emission spectra at λ_ex_ = 340 nm (Figure 5) show only Eu^3+^ emission, thus indicating a pronounced photoexcitation energy transfer from the ttdc ligand, acting as an antenna, to the highly emissive cation. The spectra for **1_DMF_**, **1_DMSO_** and **1_phet_** contain a series of strong narrow bands corresponding to the series of ^5^D_0_ → ^7^F_J_ (J = 0, 1, 2, 3, 4) phosphorescence transitions in Eu^3+^. The ^5^D_0_ → ^7^F_2_ is an electric dipole transition and, therefore, its intensity is hypersensitive to the symmetry of the Eu^3+^ coordination environment. On the other hand, ^5^D_0_ → ^7^F_1_ is a magnetic dipole transition, which is insensitive to the cation environment and the ratio between two these bands’ intensities may be used to analyze the symmetry of the local coordination environment of Eu^3+^ [47]. As shown in Table 2, all the adducts possess very high I(^5^D_0_→^7^F_2_):I(^5^D_0_ → ^7^F_1_) ratios, indicating a low symmetry of the Eu^3+^ coordination environment [48], thus matching the crystal structure data. An observed decrease in the intensity ratio, from 7.32 for **1_DMF_** down to 6.39 for **1_DMSO_**, may be attributed to the inversive disorder of the coordinated DMSO molecule (see Figure 1b), leading to some increase in the local symmetry of the metal ion. The I(^5^D_0_→^7^F_2_):I(^5^D_0_ → ^7^F_1_) for **1_phet_** possesses a value of 7.54, which is much closer to **1_DMF_**, confirming no considerable substitution of DMF in the Eu^3+^ coordination environment. For **1_cin_**, having a very low emission intensity, these bands’ ratio was estimated to be 6.95, surprisingly lower than for **1_DMF_**, as the decrease in the local symmetry of metal cation could be expected at partial substitution of DMF by cinnamal. However, in addition to the low intensity of the **1_cin_** emission spectrum and a subsequently low accuracy of the I(^5^D_0_→^7^F_2_):I(^5^D_0_ → ^7^F_1_) ratio, derived from the spectrum of the cinnamal-substituted sample, the I(total):I(^5^D_0_ → ^7^F_1_) ratio clearly increases for **1_cin_**, and this issue evidences the direct static quenching effect by cinnamal, which should result in a decrease in the I(^5^D_0_ → ^7^F_1_) relative intensity [49,50]. Further, the radiative decay lifetimes and internal quantum yields were calculated from the spectra, in terms of the Judd-Ofelt theory, using the previously described approach [49,50]. These data also show a ~25% lowering in t_rad_ for **1_cin_** and a ten times decrease (see Table 2) in its internal quantum yield compared to **1_DMF_**, illustrating the quenching impact of cinnamal.

Experimental quantum yields (QY_obs_) for **1_DMF_**, **1_DMSO_** and **1_phet_** at λ_ex_ = 340 nm were measured as 20.3(6)%, 27.4(5)% and 8.6(4)%, respectively, indicating a slight increase in the luminescence of the Eu-ttdc framework at DMF substitution by DMSO, and a considerable (more than twofold) decrease when DMF is exchanged to phenylethanal. The emission of **1_cin_** appeared to be very poor, with no detectable QY, revealing a strong quenching of the MOF luminescence upon the inclusion of cinnamal. Previously, our group reported an example of solid-state guest cinnamal sensing using the metal-organic framework, while a more than fivefold decrease in the [Tb_2_(phen)_2_(NO_3_)_2_(chdc)_2_]·2DMF solid state luminescence QY from 13.5% to 2.5% was observed [51] at MOF immersing in liquid cinnamal and further filtration. Compared to such previously described examples, the present data show very strong quenching of QY_obs_, from 20.3% to virtually 0%, which is apparently related to the direct coordination of the cinnamal molecule to the emissive center, in addition to filling the MOF channels, and a subsequent increase in non-radiative energy dissipation. The non-radiative decay provided by such a guest may have either a vibrational origin or an amplified indirect energy transfer from both Eu^3+^ and ttdc ligand excited states to Eu^3+^ ground state through an extended conjugated π-system of cinnamal [52]. Therefore, cinnamal appears to be a very strong photoexcitation quencher in such a system, effectively competing with Eu^3+^ for energy transfer.

As shown by the UV/vis absorption spectra (Appendix A), both phenylethanal and cinnamal possess low absorption in DMF at λ > 325 nm. Thus, quenching the luminescence of the red-emitting Eu(III)-based framework **1** at λ_ex_ = 340 nm indicates a considerable energy transfer from the coordination framework to phet and cin, when such π-conjugated systems act as adsorptive quenchers of the MOF luminescence with further non-radiative energy dissipation. A presence of direct, strong interactions between the Eu-ttdc and solvent moieties, along with the absorption spectra and presented unambiguous chemical characterization, is proven by the excited state lifetimes (t_l_) measurement for all the four adducts (Appendix A). As summarized in Table 2, a slight t_l_ increase of ~6% is observed for solid **1_DMSO_** compared to the starting **1_DMF_** sample. An almost threefold decrease in t_l_ appears for **1_phet_**, while the corresponding value for **1_cin_** is up to 14 times diminished compared to the DMF parent, giving an only 79 μs characteristic excited state lifetime. Such a t_l_ value is rather low for Eu^3+^, however, it still resides in the phosphorescence range of Eu(III)-based complexes [53,54,55,56,57,58]. The trends observed for the measured lifetimes and quantum yields are entirely consistent with the trends in the calculated radiation decay lifetimes and internal quantum yields (see above and Table 2). All these data, independently to single crystal structures, confirm the very close localization of the cinnamal quencher to the metal center in **1_cin_** and less effective quenching in **1_phet_** by phenylethanal, localized only in the voids of the coordination framework. On the contrary, a slight amplification of luminescence is provided by DMSO compared to DMF, possibly due to higher electron donor properties of the DMSO ligand.

Luminescence measurements for **1_DMF_** powders suspended either in pure DMF or DMF-based 1% (*v*.*v*.) solutions of DMSO, cinnamal and phenylethanal were probed. The corresponding emission spectra are shown in Figure 5b and are generally similar to the solid-state spectra, with typical to Eu^3+^ emission bands in the cases of DMF, DMSO and phet, and almost zero emission in the presence of cinnamal. The excited state lifetime (see Appendix A and Table 2) for **1_DMF_** dispersed in pure DMF is similar to the value obtained for solid **1_DMF_**, indicating the property integrity of **1_DMF_** after the filtration. t_l_ for **1** suspension in the rest of the liquid solutions are also close to these values, consistent with the obvious similarity of the main component (DMF) in all solutions. However, a slight decrease in phosphorescence lifetimes is evident in solutions containing cinnamal and phenylethanal at their ~1% volume concentrations, suggesting their adsorption into the MOF even in the diluted state.

## 3. Materials and Methods

### 3.1. Materials

Thieno[3,2b]thiophene-2,5-dicarboxylic acid (H_2_ttdc, >97.0%) was synthesized according to the previously published procedure [59]. Eu(NO_3_)_3_∙6H_2_O (99.9% REO) was received from Dalchem (Nizhny Novgorod, Russia). *N*,*N*-dimethylformamide (DMF, reagent grade) was supplied by Vekton (Saint Petersburg, Russia). Dimethylsulfoxide (DMSO, high purity grade) was received from Reaktiv (Novosibirsk, Russia). *Trans*-Cinnamaldehyde (99%) was supplied by Sigma Aldrich (St. Louis, MO, USA). Phenylethanal (98%) was received from Acros Organics (Geel, Belgium). All reagents were used as received, without further purification.

### 3.2. Instruments

Infrared (IR) spectra were obtained in a 4000−400 cm^−1^ range on a Bruker Scimitar FTS 2000 spectrometer in KBr pellets. Elemental CHNS analyses were carried out using a VarioMICROcube device. Thermogravimetric analyses (TGA) were performed on a Netzsch TG 209 F1 Iris instrument at a 10 K∙min^−1^ heating rate under inert atmosphere. Photoluminescence excitation and emission spectra and fluorescence lifetimes were recorded with a spectrofluorometer Horiba Jobin Yvon Fluorolog 3 equipped with 450W power ozone-free Xe-lamp, cooled R928/1860 PFR technologies photon detector with PC177CE-010 refrigerated chamber and double grating monochromators. Standard correction curves were used for the spectra correction for source intensity and detector response. The absolute quantum yield was measured using a G8 (GMP SA, Renens, Switzerland) spectralon-coated integrating sphere, which was connected to a Fluorolog 3 spectrofluorimeter. UV/vis absorption spectra were recorded on OKB Spectr SF-2000 spectrophotometer. Diffraction data for the single crystals of **1_DMF_, 1_DMSO_** and **1_phet_** were collected on an automated Agilent Xcalibur diffractometer, equipped with an AtlasS2 area detector and graphite monochromator (λ(MoKα) = 0.71073 Å). The CrysAlisPro program package [60] was used for the integration, absorption correction and determination of unit cell parameters. Diffraction data for the single crystals of **1_cin_** were obtained on the ‘Belok’ beamline [61,62] (λ = 0.745 Å) of the National Research Center ‘Kurchatov Institute’ (Moscow, Russian Federation) using a Rayonix SX165 CCD detector. The data were indexed, integrated and scaled, and absorption correction was applied using the XDS program package [63]. The crystallographic data and details of the structure refinements are summarized in Table A1 (see Appendix B). The dual space algorithm (SHELXT [64]) was used for structure solution and the full-matrix least squares technique (SHELXL [65]) was used for structure refinement. Anisotropic approximation was applied for all atoms, except hydrogens. The positions of the hydrogen atoms in the organic ligands were calculated geometrically and refined in the riding model. Details of the single crystal structure determination experiments and structure refinements are summarized in Table A1. CCDC 2213365–2213368 entries contain the supplementary crystallographic data for this paper. These data can be obtained free of charge from The Cambridge Crystallographic Data Center at https://www.ccdc.cam.ac.uk/structures/ (accessed on 18 November 2022).

### 3.3. Synthetic Methods

The synthesis of **1_DMF_** was carried out according to the previously published procedure [34]. The crystal structure of **1_DMF_** has been previously determined at 140 K, to contain 4 localized guest DMF molecules per [Eu_2_(DMF)_4_(ttdc)_3_] coordination framework formula unit. In the present work, the structure of **1_DMF_** was redetermined at 295 K in a glass capillary filled with DMF. No distinct guest positions were defined at the room temperature. The non-ordered electron density in the voids obtained by SQUEEZE [43] (178 electrons per 638 Å^3^) was attributed to 4.45 DMF molecules per f.u., resulting in [Eu_2_(DMF)_4_(ttdc)_3_]·4.45DMF as a final formula of the crystal.

For the syntheses of **1_solv_**, Ca. 50 mg of **1_DMF_** single crystals were immersed in 1.0 mL of the corresponding solvents, where solv = DMSO, phenylethanal (phet) or cinnamal (cin), in a glass vial. The solvent was refreshed thrice with one day interval for DMSO or with a three day interval for pheta and cin. After the immersing, the obtained solids were filtered and dried in air. Yields are close to quantitative. Single crystals suitable for SCXRD were selected for crystal structure determination before the filtration.

**1_DMSO_.** Single crystal composition: [Eu_2_(DMSO)_4_(ttdc)_3_]·2.5DMSO·2.2H_2_O. IR spectrum (KBr, cm^–1^) main bands: 3383 (s, br); 3085 (w); 3029 (w); 2929 (w); 1637 (m); 1543, 1477, 1387 and 1328 (m); 1013 (m). Elemental CHN analysis data (%): calculated for [Eu_2_(C_2_H_6_SO)_4_(C_8_H_2_S_2_O_4_)_3_]·3C_2_H_6_SO·3H_2_O: C, 28.8; H, 3.4; N, 0.0; S, 26.3. Found: C, 28.7; H, 3.8; N, 0.0; S, 25.8. TG: weight loss in the range 25…310 °C: 40%. Calculated for 7DMSO+3H_2_O: 38%.

**1_phet_.** Single crystal composition: [Eu_2_(DMF)_4_(ttdc)_3_]·3phet. IR spectrum (KBr, cm^–1^) main bands: 3390 (s, br); 3091 (w); 3029 (w); 2925 and 2863 (m); 1718 (m); 1659 (m); 1546, 1484, 1390 and 1326 (m); 1101 (m). Elemental CHN analysis data (%): calculated for Eu_2_(C_3_H_7_NO)_3_(C_8_H_2_S_2_O_4_)_3_(C_8_H_8_O)_2_(H_2_O)_2_: C, 42.8; H, 3.5; N, 2.6; S, 12.0. Found: C, 42.7; H, 3.5; N, 2.5; S, 12.2. TG: weight loss in the range 25…330 °C: 38%. Calculated for 3DMF+3phet+2H_2_O: 39%.

**1_cin_.** Single crystal composition: [Eu_2_(DMF)_3.5_(cin)_0.5_(ttdc)_3_]·1.64cin. IR spectrum (KBr, cm^–1^) main bands: 3085 (w); 2929 (w); 1673 (s); 1557, 1484, 1389 and 1323 (m). Elemental CHN analysis data (%): calculated for [Eu_2_(C_3_H_7_NO)_3.5_(C_9_H_8_O)_0.5_(C_8_H_2_S_2_O_4_)_3_]·2.5C_9_H_8_O: C, 45.2; H, 3.4; N, 3.0; S, 11.8. Found: C, 45.2; H, 3.4; N, 3.0; S, 11.9. TG: weight loss in the range 25…310 °C: 40%. Calculated for 3.5DMF+3cin: 40%.

A thin powder sample of **1_DMF_** suspended in DMF was prepared following the method described above, but with continuous intensive stirring during 48 h. After the cooling in air to room temperature, the resulting thin powder dispersion was decanted and washed with 10 mL of DMF, thrice, with the following decantation at each step to remove unreacted Eu(III) salt and H_2_ttdc. Subsequently, the suspension was diluted by DMF to 250 mL general volume, shaken manually and hit in the ultrasound bath for 5 min; then, the obtained uniform suspension was stable for at least one hour. The suspensions for the luminescent measurements were prepared by mixing 2.0 mL of the obtained **1_DMF_** dispersion with 2.0 mL of pure DMF or 2% (*v*/*v*) analyte solutions.

## 4. Conclusions

To summarize, the flexibility of a [Eu_2_(DMF)_4_(ttdc)_3_] coordination framework was successfully applied in the crystal structure determination of its new adducts with DMSO, phenylethanal and cinnamal; the two latter guests being rare in MOF host-guest chemistry. A considerable contraction of the [Eu_2_(DMF)_3.5_(cin)_0.5_(ttdc)_3_] coordination framework was observed for the adduct with cinnamal, and the sponge-like behavior of the coordination lattice allowed us to thoroughly analyze the intermolecular host-guest interactions by means of single-crystal X-ray diffraction. The solvent exchange was further investigated in terms of the luminescent response, including quantum yields (QY) and excited state lifetimes through both theoretical calculation and experimental determination. A ca. threefold decrease and an almost full quenching of Eu^3+^-based emission were observed in both QYs and the lifetimes for phenylethanal- and cinnamal- incorporated adducts, respectively. The latter was explained by direct coordination of the cinnamal strong quencher to the emissive metal cation and further non-radiative energy dissipation by such an extended organic moiety. This work provides a basis for further studies of flexible metal-organic frameworks in structure-directed luminescent sensing applications.

## Data Availability

CCDC 2213365–2213368 entries contain supplementary crystallographic data for this paper. These data can be obtained free of charge from The Cambridge Crystallographic Data Center [66].

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
