# Peer review of "Highly Luminescent Crystalline Sponge: Sensing Properties and Direct X-ray Visualization of the Substrates"

_molecules, 2022, doi:10.3390/molecules27228055_

Round 1

Reviewer 1 Report

In this paper, the authors selected a previous reported MOF with potential structural dynamics and luminescent sensing abilities to be the host compound, and investigated the crystal structures and luminescent properties of solvent-exchanged samples by the crystalline sponge method. Three exchanged compounds were obtained, and their structures were determined by single-crystal XRD and other technologies. Luminescent properties of these compounds were also investigated. Overall, this work is well-organized and the characterizations for the prepared materials are systematic and adequate. I recommend its publication on Molecules after the following questions are well addressed.

1.     The Introduction part is kind of short. Examples of how to apply a crystalline sponge method in structural analysis of flexible MOFs should be added.

2.     “coordination number” should show before “CN” in line 66 and 70, Page 2.

3.     The abbreviations for metal and ions should be in a uniform layout, such as “Eu3+”, “Eu(III)”, “lanthanide(III)”in line 70-72, page 2.

4.     In table 1, the parallelogram angles and void volume for 1DMF measured at 140 K and 295K are both the same, why V/Z is different? What temperature is applied for the structure analysis of other exchanged compounds?

5.     In figure 3a, the ligands are not shown properly. So does that in figure S1b. If these present the disorder of ligands, then figure 3b should be revised.

6.     The crystallographic data for this paper is not found in CCDC https://www.ccdc.cam.ac.uk/structures/. The checkcif documents may be provided to the reviewers for reference. 

Author Response

Reviewer 1

In this paper, the authors selected a previous reported MOF with potential structural dynamics and luminescent sensing abilities to be the host compound, and investigated the crystal structures and luminescent properties of solvent-exchanged samples by the crystalline sponge method. Three exchanged compounds were obtained, and their structures were determined by single-crystal XRD and other technologies. Luminescent properties of these compounds were also investigated. Overall, this work is well-organized and the characterizations for the prepared materials are systematic and adequate. I recommend its publication on Molecules after the following questions are well addressed.

Authors thank the reviewer for high evaluation of the work and helpful suggestions for the article improvement.  

  1. The Introduction part is kind of short. Examples of how to apply a crystalline sponge method in structural analysis of flexible MOFs should be added.

Following the reviewer’s suggestion, the introduction has been expanded:

For example, up to four times variation of unit cell parameters has been reported for MIL-88 series upon activation or inclusion of water and light alcohols [22,23]. In-situ X-ray diffraction methods have been successfully applied in the structure determination of three [24] or five [25] forms of MOF crystalline sponge upon single gas adsorption.

The reference list was expanded in the revised manuscript accordingly.

  1. “coordination number” should show before “CN” in line 66 and 70, Page 2.

Done.

  1. The abbreviations for metal and ions should be in a uniform layout, such as “Eu3+”, “Eu(III)”, “lanthanide(III)”in line 70-72, page 2.

Done.

  1. In table 1, the parallelogram angles and void volume for 1DMFmeasured at 140 K and 295K are both the same, why V/Z is different? What temperature is applied for the structure analysis of other exchanged compounds?

An observed difference in V/Z for the structure of 1DMF determined at two temperatures is apparently attributed to the slight thermal expansion of the coordination lattice. Rounded to integer values parallelogram angles for these structures are indeed the same, but two of three parallelogram edges (i.e. distances between binuclear metal block centroids) are larger for the structure determined at higher temperature, reasonably resulting in a slight increase in its unit cell volume. These edge lengths are shown below:  

1DMF at 140 K: 11.92 Å; 13.04 Å; 15.85 Å.

1DMF at 295 K: 12.02 Å; 13.01 Å; 15.88 Å.

For convenience, Figure 1d was also revised to contain initial ttdc ligands shown with 70 % transparency. We believe it would make the schematic illustration of the parallelogram clearer.

For 1DMSO and 1phet adducts, the single-crystal XRD experiment temperature was 150 K. Crystal structure of 1cin was determined at 100 K. These data are summarized in Table A1.

  1. In figure 3a, the ligands are not shown properly. So does that in figure S1b. If these present the disorder of ligands, then figure 3b should be revised.

We highly appreciate such a careful observation of the reviewer. For better clarity of the figures, we decided to clear the second positions of disordered ttdc heterocyclic cores from Fig. 3a and S1b. The necessary comments have been added into the figure captions.

  1. The crystallographic data for this paper is not found in CCDC https://www.ccdc.cam.ac.uk/structures/. The checkcif documents may be provided to the reviewers for reference. 

The unpublished structures are to be provided by CCDC team upon a request. Please, insert the CCDC entry ID on https://www.ccdc.cam.ac.uk/structures/ and then fill the interactive form as a reviewer (check the corresponding box). For a convenience, the checkcif files are added to the submission in the “unpublished data” section.

Reviewer 2 Report

The manuscript entitled ‘Highly luminescent crystalline sponge: sensing properties and direct X-ray visualization of the substrates’ presents a study on an Eu(III) metal-organic framework, which acts as a crystalline sponge towards small organic molecules. It describes three new adducts of the host (with DMSO, phenylethanal, trans-cinnamaldehyde) and their crystal structures, as well as the results of luminescence studies performed on the obtained materials. The manuscript is interesting though in two of the presented cases, the authors did not manage to fully model the electron density present in the channels (DMSO, trans-cinnamaldehyde), which is necessary for any proper structural analyses treating about host-guest interactions and for performing any further studies on such materials. Most likely, the studied materials are not homogeneous and contain crystals with different amounts of guests/water molecules. I would recommend the manuscript for publication after the authors have optimised the soaking step of these experiments (by extending the time or increasing the temperature of the process), which would lead to enhanced site occupancy of the guest molecules. Otherwise, it is difficult to draw any meaningful conclusions.

Author Response

Please find the attached pdf file. 

Reviewer 3 Report

The paper by Demakov et al. describes crystal structures and luminescent properties of three novel adducts of a flexible Eu-based porous three-dimensional metal-organic framework. The study is well-designed, and the data support the conclusions. The study may provide a basis for further exploration of flexible metal-organic frameworks in structure-directed luminescent sensing applications.
The paper is exciting, well-written, original, and within the scope of the Journal. I recommend its acceptance in the present form.

Author Response

Authors thank the reviewer for such high evaluation of the work. However, several corrections have been done according to the notes suggested by other reviewers.

Reviewer 4 Report

In this work, authors describe a series of adducts of a porous three-dimensional metal-organic framework [Eu2(DMF)4(ttdc)3] with different guest molecules (DMF, DMSO, phenylethanal, cinnamaldehyde). The adducts were unambiguously characterized by complementary techniques so that the crystallographic positions of the adsorbed molecules were even successfully determined by X-ray diffraction analysis. Photophysical study of the above adducts revealed that the inclusion of DMSO into parent MOF results in a slight increase in PL quantum yield, while the uptake of phenylethanal leads to a considerable PL decrease. The most pronounced quenching effect was observed for cinnamal, incorporation of that make the parent MOF completely non-emissive. Overall, the reviewed work is sound, well-written, and its results noticeably contribute to the MOF chemistry field. There is no doubt that this work will be of interest to the readership of this journal. Thus, I recommend this manuscript to be published as it. However, after minor revision to address the following issues. I would like to leave the following comments that may be helpful in improving the presentation of the article:

1.         It is desirable to provide the excitation spectra in order to understand why 340 nm light was used as excitation source.

2.         For the best visual clarity, it is recommended to add labels for selected atoms (Eu, S, N) in figures 1 and 2.

3.         I would like to draw the attention of the authors to the discussion in article DOI 10.1039/C8RA09285K, which will probably be useful for a deeper understanding of the PL quenching by cinnamic aldehyde.

Author Response

Reviewer 4

In this work, authors describe a series of adducts of a porous three-dimensional metal-organic framework [Eu2(DMF)4(ttdc)3] with different guest molecules (DMF, DMSO, phenylethanal, cinnamaldehyde). The adducts were unambiguously characterized by complementary techniques so that the crystallographic positions of the adsorbed molecules were even successfully determined by X-ray diffraction analysis. Photophysical study of the above adducts revealed that the inclusion of DMSO into parent MOF results in a slight increase in PL quantum yield, while the uptake of phenylethanal leads to a considerable PL decrease. The most pronounced quenching effect was observed for cinnamal, incorporation of that make the parent MOF completely non-emissive. Overall, the reviewed work is sound, well-written, and its results noticeably contribute to the MOF chemistry field. There is no doubt that this work will be of interest to the readership of this journal. Thus, I recommend this manuscript to be published as it. However, after minor revision to address the following issues. I would like to leave the following comments that may be helpful in improving the presentation of the article:

Authors thank the reviewer for such high evaluation of the work and helpful suggestions for the article improvement. 

  1. It is desirable to provide the excitation spectra in order to understand why 340 nm light was used as excitation source.

Excitation spectra are now added into ESI as Fig. S6. 340 nm wavelength light was chosen as giving the softest excitation energy prior to the excitation absorption edges for 1solv. Also, 340 nm wavelength does not interfere with the strong absorption region for phenylethanal and cinnamal (Fig. S7).

  1. For the best visual clarity, it is recommended to add labels for selected atoms (Eu, S, N) in figures 1 and 2.

Following the reviewer’s suggestion, Figure 1 was appended by color legend. We decided not to add the legend to figure 2 to avoid its overloading. A comment on color designations is presented in figure caption.

  1. I would like to draw the attention of the authors to the discussion in article DOI 10.1039/C8RA09285K, which will probably be useful for a deeper understanding of the PL quenching by cinnamic aldehyde.

Authors thank the reviewer for the valuable note. The text in lines 255-260 has been rewritten to include an additional possible mechanism of the luminescence quenching by cinnamal, discussed in that paper:

The non-radiative decay provided by such guest may have either vibrational origin or amplified indirect energy transfer from both Eu3+ and ttdc ligand excited states to Eu3+ ground state through an extended conjugated π-system of cinnamal [10.1039/C8RA09285K]. Therefore, cinnamal appears to be a very strong photoexcitation quencher in such system, effectively competing with Eu3+ for energy transfer.
